

# Determination of zenith hydrostatic delays and the development of new global long-term GNSS-derived precipitable water vapor

Xiaoming Wang[1], Kefei Zhang[1, 2], Suqin Wu[1], Changyong He[1], Yingyan Cheng[3], and Xingxing Li[4]

[1]Satellite Positioning for Atmosphere, Climate and Environment (SPACE) Research Centre, the School of Science, Mathematical and Geospatial Sciences, RMIT University, AUSTRALIA
[2]China University of Mining and Technology, CHINA.
[3]Chinese Academy of Surveying and Mapping, Beijing, CHINA
[4]Helmholtz-Zentrum Potsdam – Deutsches GeoForschungsZentrum GFZ, Brandenburg, GERMANY

*Correspondence to*: Kefei Zhang (kefei.zhang@rmit.edu.au)

**Abstract.** Surface pressure is a vital meteorological variable for the accurate determination of precipitable water vapor (PWV) using Global Navigation Satellite Systems (GNSS). The lack of pressure observations is a big issue for the study of climate using historical GNSS observations, which is a relatively new area of GNSS applications in climatology. Hence the use of the surface pressure derived from either an empirical model (e.g. Global Pressure and Temperature 2 wet, GPT2w) or a global atmospheric reanalysis (e.g. ERA-Interim) becomes an important alternative solution. In this study, pressure derived from these two methods is compared against the pressure observed at 108 global GNSS stations for the period 2000–2013. Results show that a good accuracy is achieved from the GPT2w-derived pressure in the latitude band of $-30^0\sim30^0$ and the average value of Root-Mean-Square (RMS) errors across all the stations in this region is 2.4 mb. Correspondingly, an error of 5.6 mm and 1.0 mm in its resultant zenith hydrostatic delay (ZHD) and PWV is expected. In addition, GPT2w-derived pressure usually has a larger error in the cold season due to large diurnal ranges, which is not considered in the GPT2w model. The average value of the RMS errors of the ERA-Interim-derived pressure across all the 108 stations is 1.1 mb, which will lead to an equivalent error of 2.5 mm and 0.4 mm in its resultant ZHD and PWV respectively. Our research also indicates that the ERA-Interim-derived pressure has the potential to be used as a useful meteorological data source to obtain high accuracy PWV on a global scale for climate studies and the GPT2w-derived pressure can be potentially used for climatology as well although it may be only suitable for the tropical regions.

**Keywords:** GNSS, surface pressure, ZHD, PWV, ERA-Interim, GPT2w

## 1. Introduction

Water vapor (WV) as a principal atmospheric parameter is a central component in both earth's energy budget and water cycle. Accurate knowledge of WV is not only vital for weather forecasting but also an important independent data source for global climate studies. For the last decade, Global Navigation Satellite Systems (GNSS) have been used as an emerging and





robust technique for remotely sensing precipitable WV (PWV) for the monitoring of the real-time PWV variations in the atmosphere (Schneider et al., 2010;Rohm et al., 2014;Zhang et al., 2015;Li et al., 2014;Li et al., 2015;Guerova et al., 2016) or the studies of climate (Nilsson and Elgered, 2008;Jin and Luo, 2009;Vonder Haar et al., 2012;Ning and Elgered, 2012;Alshawaf et al., 2016) due to their 24-hour availability, high accuracy, global coverage, high resolution and low cost. The atmospheric parameter directly estimated from GNSS measurements is the GNSS signal's tropospheric zenith total

delay (ZTD) which can be effectively divided into the zenith hydrostatic delay (ZHD) and the zenith wet delay (ZWD). The ZHD can be accurately determined using the surface pressure observed by meteorological sensors. The GNSS-derived PWV over a station can then be obtained by multiplying the ZWD with a conversion factor $\Pi$ which is a function of water-vapor-weighted mean temperature $T_m$ over the station. The $T_m$ can be determined using one of the following three methods: 1) temperature and humidity profiles from either radiosonde observations or atmospheric reanalysis datasets (Wang et al.,

2005;Wang et al., 2016); 2) the relationship between surface temperature $T_s$ and the water-vapor-weighted mean temperature $T_m$ (Bevis et al., 1992;Bevis et al., 1994;Ross and Rosenfeld, 1997); and 3) an empirical model developed from atmospheric reanalysis products (Lagler et al., 2013;Böhm et al., 2014;Yao et al., 2012;Yao et al., 2015).

Motivated by our early research (Wang et al., 2016), it is vital to assess the performance of different methods for determining the ZHD on a global scale, which is essential in the development of a reliable global long-term PWV time series

for climate studies. Although the ZHD can be accurately obtained from surface pressure observations, few GNSS stations were installed with meteorological sensors back in the 1990s since these stations were established mainly for precise positioning and navigation applications. Therefore, the lack of meteorological data (i.e. pressure) at these stations is a serious issue for the use of these historical GNSS data for global climate studies. To address this issue, an alternative method is to use pressure derived from a global atmospheric reanalysis (e.g. ERA-Interim) or an empirical model (e.g. Global Pressure

and Temperature 2 wet, GPT2w). In this study, the errors in the pressure derived from these two approaches are investigated and the impact of these errors on the subsequent ZHD and PWV determination has also been studied. The global atmospheric reanalysis datasets used are the ERA-Interim which is produced by European Centre for Medium-Range Weather Forecasts (ECMWF) (Dee et al., 2011) and has been widely used for global climate studies. The empirical model selected is Global Pressure and Temperature 2 wet (GPT2w) (Böhm et al., 2014), which provides the mean value plus annual

and semiannual amplitudes of the pressure.

For the performance assessment of these two methods, the ERA-Interim-derived pressure (using two computation methods) and the GPT2w-derived pressure are compared against the surface pressure measured at the 108 global GNSS stations during the period 2000–2013. Then the impact of the error in these pressure values on the ZHD is evaluated by comparing the resultant ZHD against the ZHD derived from the surface pressure measured. Similarly, the impact of these errors on the

PWV is also assessed. In the determination of PWV, the ZTD reanalysis products provided by the Center for Orbit Determination in Europe (CODE), and the conversion factor $\Pi$ determined using the humidity and temperature profiles from the ERA-Interim are used.



## 2.    Datasets

Recently, re-processing of historical GNSS data has emerged as a new initiative in the world GNSS community driven by the high demand of various new scientific studies. For example, the CODE has reprocessed GNSS observations at 371 global IGS (International GNSS services) stations for the period of 1994–2013 and made the results public to the science community (see: ftp://ftp.unibe.ch/aiub/REPRO_2013/). Some of these centers have also collected meteorological observations at these IGS stations e.g. the Scripps Orbit and Permanent Array Center (SOPAC) (see: http://garner.ucsd.edu/pub/met/). In this study, the meteorological observations provided by the SOPAC and the ZTD results provided by the CODE are used to determine ZHD and PWV.

Since the time series of both the GNSS-derived ZTDs and the pressure measured contain gaps, the data missing rates of these time series need to be investigated. As a result, a comparison between the meteorological observations and ZTDs over 371 global stations at four epochs (00, 06, 12, 18 UTC) of each day during the whole period of 2000–2013 is carried out. The missing rates of pressure observations compared to the ZTDs are shown in Fig. 1, which indicates that at most stations only few meteorological observations were recorded. Figure 2 indicates that the missing rate smaller than 25 % happens only at 6 % of the stations and there were no pressure observations recorded at 67 % of the stations. Thus, a reliable alternative method for obtaining surface pressure is needed for the determination of PWV from historical GNSS observations.

As mentioned in the previous section, the ERA-Interim is also used to determine the conversion factor Π and compute surface pressure. The ERA-Interim data cover the period from 1 January 1979 onwards and near real-time information is also generated. All these are available at 00, 06, 12, 18 UTC of each day. Its spatial resolution is approximately 80 km in horizontal and at 60 vertical levels from the surface up to 0.1 hpa.

## 3.    Methods for determining pressure from ERA-Interim

More details on the determination of the pressure from GPT2w can be found in (Böhm et al., 2014). The main focus of this section is the details of two computation methods used to obtain the pressure over GNSS stations from the ERA-Interim.

### 3.1    One-point method

The one-point method uses meteorological data at an ERA-Interim grid that is closest to the GNSS station to compute the pressure $P_s$ for the station (Bosy et al., 2010;Karabatić et al., 2011;Wilgan et al., 2015) by,

$$P_s = p_c \left[ \frac{T_c - \gamma(H_s - H_c)}{T_c} \right]^{\frac{g \cdot M}{R \cdot \gamma}} \tag{1}$$

where $p_c$, $T_c$ and $H_c$ are the air pressure, temperature (in kelvins) and height (in meters) of the nearest ERA-Interim grid point respectively, $\gamma$=0.0065 K m$^{-1}$ is the standard temperature lapse rate, $H_s$ is the station height, $M$=0.0289644 kg mol$^{-1}$ is a molar mass of dry air, $R$=8.31432 [N·m (mol·K)$^{-1}$] is an ideal gas constant, and $g$ is a gravitational parameter which can be determined by,





$$g = 9.8063 \cdot \left\{ 1 - 10^{-7} \frac{H_c + H_s}{2} [1 - 0.0026373 \cdot cos(2\varphi) + 5.9 \cdot 10^{-6} \cdot cos^2(2\varphi)] \right\} \tag{2}$$

where $\varphi$ is the latitude of the GNSS station.

## 3.2    Four-point method

The four-point method (Schüler, 2001) uses the pressure values from two nearest pressure levels at four neighboring grid points to determine the pressure for the GNSS station $(\varphi_s, \lambda_s, H_s)$. Two procedures need to be carried out: a vertical computation procedure and a horizontal computation procedure. The vertical computation uses the pressure values at these two pressure levels to determine the pressure at the height of the GNSS station at four nearby grid points. Let the horizontal coordinates of the four grids be $(\varphi_i, \lambda_i), i = 1, 2, 3, 4$, the heights of the two nearest pressure levels at the $i^{th}$ grid be $H_i^1$ and $H_i^2$, and the pressure values at these two levels be $p^1$ and $p^2$, respectively. The pressure value $p_i$ at the $i^{th}$ grid and the height of the GNSS station (i.e. $H_s$) can then be determined by,

$$p_i = \frac{w_i^1}{w_i^1 + w_i^2} p_i^1 + \frac{w_i^2}{w_i^1 + w_i^2} p_i^2 \tag{3}$$

where $p_i^1$ and $p_i^2$ are the two pressure values at the height of $H_s$ at the $i^{th}$ grid calculated using $p^1$ and $p^2$, respectively, by,

$$p_i^1 = p^1 \left[ 1 + \left( \frac{8.419 \times 10^{-5} \left( H_i^1 - H_s \right)}{(p^1)^{0.190284}} \right) \right]^{5.255303} \tag{4}$$

$$p_i^2 = p^2 \left[ 1 + \left( \frac{8.419 \times 10^{-5} \left( H_i^2 - H_s \right)}{(p^2)^{0.190284}} \right) \right]^{5.255303} \tag{5}$$

and $w_i^1$ and $w_i^2$ in Eq. (3) are two weighting coefficients calculated by,

$$w_i^1 = \frac{1}{\left( H_i^1 - H_s \right)^2} \tag{6}$$

$$w_i^2 = \frac{1}{\left( H_i^2 - H_s \right)^2} \tag{7}$$

After all the pressures $p_i$ $(i = 1, 2, 3, 4)$ in Eq. (3) are obtained, the next step is to horizontally interpolate the pressure value at the GNSS station $(\varphi_s, \lambda_s, H_s)$ by,

$$p_s = \sum_{i=1}^{4} \overline{w_i} p_i \tag{8}$$

where $\overline{w_i}$ is a weighting coefficient computed by,

$$\overline{w_i} = \frac{w_{*i}}{w_{*1} + w_{*2} + w_{*3} + w_{*4}} \quad (i = 1, 2, 3, 4) \tag{9}$$

where $w_{*i}$ is a weighting coefficient for the $i^{th}$ grid point calculated by,

$$w_{*i} = \psi_i^{-C} \tag{10}$$

where $\psi_i$ is the spherical distance between the grid point and the GNSS station and $C$ is the weighting power. As suggested by Schüler (2001), $C$ can be set to a value between 1 and 1.5 for the pressure computation. In this study, comparisons among the pressures obtained from $C = 1.0, 1.1, 1.2, 1.3, 1.4$, and $1.5$ at all the 108 stations for the period 2000–2013 are conducted and the results indicate that the largest difference is between 1.0 and 1.5. The mean of the RMS values of the differences





between these two sets of pressure values across all these 108 stations is about 0.05 millibar (mb), which will introduce a difference of only 0.12 mm in the resultant ZHD. In this study, $C$ is set to 1.0 since the difference from different C values is very small.

## 4. Comparison and analysis

The pressure and ZHD at 108 stations for the period 2000–2013 determined from both GPT2w and the ERA-Interim using the aforementioned two computation methods are compared against the surface pressure measurements and their resultant ZHD. The comparison results are analysed in Sect. 4.1 and 4.2. Since the ERA-Interim-derived pressure and ZHD are of very high accuracy on a global scale, the pressure and ZHD over 371 global stations at four epochs (00, 06, 12, 18 UTC) of each day during the whole period of 2000–2013 are derived from the ERA-interim and subsequently used to analyse their temporal variations. The analyses results are shown in Sect. 4.3. Section 4.4 shows the comparisons among the PWVs computed using the aforementioned four sets of pressure.

### 4.1 Comparisons of pressure and ZHD in spatial domain

#### 4.1.1 Comparisons of pressure in spatial domain

To investigate the performance of the GPT2w and ERA-Interim methods, the bias and RMS of the resultant pressures at 108 stations for the period of 2010–2013 (using observed pressure as a reference) are computed. The pressure derived from the GPT2w model, the ERA-Interim with the one-point method, and the ERA-Interim with the four-point method are named as *P_GPT2w*, *P_ERA1* and *P_ERA4*, respectively. As shown in Fig.3, the biases of *P_GPT2w* (Fig. 3 (a1)), *P_ERA1* (Fig. 3 (a2)), *P_ERA4* (Fig. 3 (a3)) are between −1 and 1 mb at most stations. No obvious correlation between the magnitudes of the bias and the latitude is found. However, the RMS error of *P_GPT2w* (Fig. 3 (b1)) is very latitude dependent and also larger than that of both *P_ERA1* (Fig. 3 (b2)) and *P_ERA4* (Fig. 3 (b3)) at the same stations. For the 28 stations located in the low-latitude band of −30°~30°, the RMS errors of *P_GPT2w* are in the range of 1.0~4.3 mb and the mean of these 28 RMS errors is 2.4 mb. For the 65 stations located in the mid-latitude bands of −30°~−60° and 30°~60°, the RMS errors of *P_GPT2w* are in the range of 2.8~12.0 mb with the mean of 7.4 mb. For the stations located in the high-latitude belts of −60°~−90° and 60°~90°, the RMS errors are within 7.6–12.6 mb and the mean value reaches 9.8 mb. Therefore, GPT2w has a far better performance in the tropical regions than the other regions in the determination of surface pressure. As shown in Fig. 3 (b2) and Fig. 3 (b3), the results from the ERA-Interim using the two aforementioned computation methods have a similar accuracy. The RMS errors of both *P_ERA1* and *P_ERA4* are in the range of 0.2~4.0 mb and the mean values of these two sets of RMS errors are both to be 1.1 mb. The percentages of those RMS errors of *P_ERA1* and *P_ERA4* across the 108 stations less than 1.5 mb reach 84 % and 80 %, respectively.

### 4.2 Comparisons of ZHD in spatial domain





To assess the impact of the errors in the three pressure sets, i.e. *P_GPT2w*, *P_ERA1*, and *P_ERA4*, on their resultant ZHD, the biases and RMS errors of the ZHDs with respect to ZHD values derived from the observed surface pressure are calculated. One of the most commonly used methods to obtain the ZHD is using the Saastamoinen formula, which is a function of surface pressure (Saastamoinen, 1972;Elgered et al., 1991;Niell et al., 2001). Davis et al. (1985) pointed out that

the uncertainty in the ZHD obtained from the Saatamoinen formula was 0.5 mm, if uncertainties in the physical constants and the calculation of the mean value of gravity were taken into consideration. This magnitude of uncertainty will introduce an uncertainty of less than 0.1 mm in its subsequent PWV determination, which can be neglected.

Equations (11) and (12), developed by Elgered et al. (1991) based on the Saastamoinen formula, is adopted in this study to obtain the ZHD (in millimeters). These formulae have been widely used in many studies (Bevis et al., 1992;Bock and

Doerflinger, 2001;Bokoye et al., 2003;Kleijer, 2004;Musa et al., 2011;Kumar et al., 2013;Norazmi et al., 2015).

$$ZHD = (2.779 \pm 0.0024)P_s/f(\varphi, H) \tag{11}$$

where $P_s$ is the total pressure (in mb) at the station's height (in kilometers) and

$$f(\varphi, H) = 1 - 0.00266 \cdot cos(2\varphi) - 0.00028H \tag{12}$$

accounts for the variation in the gravitational acceleration at the station with latitude $\varphi$ and height $H$ above a reference

ellipsoid.

For convenience, ZHDs resulting from *P_GPT2w*, *P_ERA1* and *P_ERA4* are named as *ZHD_GPT2w*, *ZHD_ERA1*, and *ZHD_ERA4*, respectively. As indicated in Fig. 4, the biases of *ZHD_GPT2w* (Fig. 4 (a1)), *ZHD_ERA1* (Fig. 4 (a2)), and *ZHD_ERA4* (Fig. 4 (a3)) are in the range of –3~3 mm at most stations. The RMS errors of *ZHD_GPT2w* (Fig. 4 (b1)) are noticeably larger than that of both *ZHD_ERA1* (Fig. 4 (b2)) and *ZHD_ERA4* (Fig. 4 (b3)). Since the RMS errors of

*P_GPT2w* are latitude dependent, the RMS errors of its resultant ZHD i.e. *ZHD_GPT2w* also show a strong dependence on latitudes. For the stations located in the low-latitude belts, the RMS errors of *ZHD_GPT2w* are within 2.3–9.9 mm with a mean of 5.6 mm. For the stations located in the mid-latitude belts, the RMS errors of *ZHD_GPT2w* are in the range of 6.6–27.1 mm with a mean value of 16.8 mm. For the stations located in the high-latitude belts, the RMS errors of *ZHD_GPT2w* are in the range of 17.4–28.7 mm with a mean of 22.3 mm. As shown in Fig. 4 (b2) and Fig. 4 (b3), the RMS errors of

*ZHD_ERA1* and *ZHD_ERA4* of the same station are similar and all in the range of 0.5–9.1 mm and the mean values of these two sets of RMS errors are about 2.5 mm. The percentages of the RMS errors of *ZHD_ERA1* and *ZHD_ERA4* at all the 108 stations less than 3 mm reach 78 % and 72 %, respectively. As suggested by previous studies (e.g. (Rohm and Bosy, 2007;Wilgan et al., 2015)), surface pressure at mountainous areas usually experiences rapid and frequent changes. The RMS errors of the *ZHD_ERA1* and *ZHD_ERA4* calculated at station WROC located in the Sudety Mountains (Poland) are 2.2 mm

and 2.4 mm respectively. This indicates that the ERA-Interim can be used in the mountainous areas for the determination of ZHD and the two methods used in this study perform in a very similar manner.

## 4.3     Comparisons of pressure and ZHD in temporal domain





In this section, annual amplitudes and diurnal ranges of pressure and ZHD over 371 stations for the period 2000–2013 derived from ERA-interim are investigated. Then the temporal characteristics of the errors in the pressure and ZHD derived

from the aforementioned three methods, especially from GPT2w, are studied. Figure 5 shows the mean values and the annual amplitudes of ERA-Interim-derived pressure and ZHD at 371 GNSS stations for the period of 2000–2013. As shown in the Fig. 5 (a1) and (a2), as expected, the mean values of pressure and ZHD at a GNSS station are closely related to the station height. However, as indicated by Fig. 5 (b1) and (b2), the annual amplitudes of the pressure and ZHD do not show a very strong dependence on the station height, but vary by locations. For example, the amplitudes seem larger over stations in

Eastern Asia than that in Western Europe. The semi-annual amplitudes are much smaller than the annual amplitudes and thus are not discussed here. Apart from annual and semi-annual variations, obvious diurnal ranges, which are defined as the difference between the maximum and minimum values at the same day, are also found at most stations. As indicated by Fig. 6, the diurnal range of pressure and ZHD are related not only to the location of a station but also to the seasons. For example, the diurnal range in the northern polar region is obviously larger during the months from December to February than those

from June to August. However, in the southern Polar Regions, the diurnal range is larger during June to August than that from December to February. It should be noted that, in the northern hemisphere the cold season (i.e. winter) is from December to February, whilst in the southern hemisphere the cold season is from June to August. Therefore, the diurnal range is usually larger in the cold season than that in the warm season. In addition, the differences between the diurnal ranges in cold and warm seasons are very small across the stations in the Tropics.

Since the GPT2w model is built based on the monthly mean pressure level data of ERA-Interim and only takes into consideration the annual and semi-annual variations, it cannot reflect the diurnal variations of both pressure and ZHD. Therefore, theoretically, the errors in the GPT2w-derived pressure and ZHD should be larger in the cold season than that in the warm season. This phenomenon was reaffirmed in our study. Figs. 7 and 8 show the errors of the pressure and ZHD derived from the aforementioned three methods at the station SUTM (32.8⁰ S, 20.8⁰ E) located in the southern hemisphere

and the station JPLM (34.2⁰ N, 118.2⁰ W) in the northern hemisphere, respectively. As shown in Fig. 6, the errors of the $P\_GPT2w$ and $ZHD\_GPT2w$ at SUTM in the southern hemisphere are obviously larger in June–August (winter in the southern hemisphere) than that in December–February. In contrast, for JPLM (Fig. 8) located in the northern hemisphere the errors of $P\_GPT2w$ and $ZHD\_GPT2w$ are noticeably larger in December–February (winter in the northern hemisphere) than that in June–August.

Fig. 9 shows the RMS errors of the GPT2w-derived ZHD at 81 GNSS stations with a time span of more than 3 years and the missing rate is smaller than 50 %. It should be noted that for long-term climate study, the length of the PWV data used should be as long as possible (e.g. >10 years). The results indicate that the magnitudes of these errors are usually larger in the cold season than that in the warm season.

## 4.4    Comparisons of PWV





Similarly to Sect. 4.2, the impact of the errors in the above three sets of pressure values on their resultant PWV is also analysed using the biases and RMS errors of the PWV. In the calculation of the PWV with Eq. (13), the ZTD provided by CODE and the conversion factor $\Pi$ computed from the temperature and humidity profiles from the ERA-Interim are adopted.

$$PWV = \Pi \cdot ZWD = \Pi \cdot (ZTD - ZHD) \tag{13}$$

More details about the computation of $\Pi$ have been published in (Wang et al., 2016) and here is only a brief description of

the calculation of $\Pi$ is given. As a function of $T_m$, $\Pi$ can be calculated using,

$$\Pi = \frac{10^6}{\rho R_v \left[\frac{k_3}{T_m} + k_2'\right]} \tag{14}$$

$$k_2' = k_2 - mk_1 \tag{15}$$

where $\rho$ is the density of liquid water, $R_v$ is the specific gas constant for water vapor, and $m$ is the ratio of the molar masses of WV and dry air. The values of physical constants $k_1 = 70.60 \pm 0.05\ K\ mb^{-1}$, $k_2 = 70.4 \pm 2.2\ K\ mb^{-1}$, and $k_3 =$

$3.739 \pm 0.0012\ 10^5\ K^2\ mb^{-1}$ are from the widely used formula for atmospheric refractivity (Bevis et al., 1994) and the constant $k_2'$ derived from Eq. (15) was set to $22.1 \pm 2.2\ K\ mb^{-1}$ as suggested by Bevis et al. (1994).

$T_m$ in Eq. (14) is water-vapor-weighted mean temperature (Davis et al., 1985) and approximated as,

$$T_m = \frac{\int \frac{P_v}{T} dz}{\int \frac{P_v}{T^2} dz} \approx \frac{\Sigma_{i=1}^{N} \frac{P_{vi}}{T_i} \Delta z_i}{\Sigma_{i=1}^{N} \frac{P_{vi}}{T_i^2} \Delta z_i} \tag{16}$$

where, $P_v$ is the partial pressure (in hPa) of WV, $T$ is the atmospheric temperature (in kelvin), and $i$ is the $i^{th}$ pressure level.

$P_v$ can be calculated using,

$$P_{si} = 6.11 \times 10^{\left(\frac{7.5 \times Td_i}{237.3 + Td_i}\right)} \tag{17}$$

$$P_{vi} = \frac{rh_i \cdot P_{si}}{100} \tag{18}$$

where $P_s$ is the saturated vapor pressure, $rh$ is the relative humidity $Td$ is the atmospheric temperature (in Celsius).

### 4.4.1 Comparison of PWV in spatial domain

The errors in the PWVs are investigated in spatial domain to study their spatial characteristics. PWVs obtained from surface pressure *P_GPT2w*, *P_ERA1*, and *P_ERA4* are named as *PWV_GPT2w*, *PWV_ERA1*, and *PWV_ERA4*, respectively. As indicated in Fig. 10, the biases of *PWV_GPT2w* (Fig. 10 (a1)), *PWV_ERA1* (Fig. 10 (a2)), and *PWV_ERA4* (Fig. 10 (a3)) are all in the range of –0.5~0.5 mm at most stations. The RMS errors of *PWV_GPT2w* (Fig. 10 (b1)) are obviously smaller in low-latitude regions than that in high-latitude regions. For all the stations located in the low-latitude belts, the RMS errors of

*PWV_GPT2w* are within 0.5~1.6 mm and the mean value of these RMS errors is 1.0 mm. For all the stations located in the mid-latitude belts, the RMS errors of *PWV_GPT2w* are in the range of 1.1~4.2 mm with the mean of 2.6 mm. For all the stations located in the high-latitude belts, the RMS errors of *PWV_GPT2w* are in the range of 2.6–4.2 mm with the mean of 3.4 mm. The RMS errors of *PWV_ERA1* and *PWV_ERA4* of the same stations are similar and all in the range of 0.1~1.5 mm with the mean of about 0.4 mm.



As suggested by The E-GVAP (http://egvap.dmi.dk/), the "breakthrough" accuracy of PWV for climate study should be 1.5 mm; the "goal" accuracy of PWV for climate study should be around 1 mm. As defined by the E-GVAP, the "goal" accuracy is an ideal requirement and no further improvements are needed if this requirement can be met. The "breakthrough" accuracy, if achieved, would result in a significant improvement for the climate study and may be considered as an optimum, from a cost-benefit point of view, when planning or designing observing systems. The statistic results of the RMS errors of

*PWV_GPT2w* show that the errors are larger than 1.5 mm at about 91 % (73 of 80) stations located within the mid/high-latitude belts. Moreover, this is only the error in the PWV caused by the error in the pressure i.e. the errors in the GNSS-estimated ZTD and the conversion factor Π are not taken into account. Therefore, for the GPT2w, it cannot be used in the the mid to high latitude belts for computing the PWV with a "breakthrough" accuracy for the climate studies. On the contrary, the ERA-Interim can be used as a potential useful source of data for computing the PWV for the climate studies on a global

scale since the percentages of the RMS errors of *PWV_ERA1* and *PWV_ERA4* at all the 108 stations that are less than 0.5 mm reach 78 % and 75 %, respectively.

## 5.     Conclusions

The lack of meteorological data makes it very difficult to take the full advantage of historical GNSS data for climate studies. This research is our continuous effort to extend our research presented in (Wang et al., 2016) through investigating the

alternative methods for the determination of ZHD and developing of a new global long-term PWV time series, which is critical for an improved understanding of climate change using the state of the art GNSS technology. Our new PWV dataset across 371 stations for the period of 2000–2013 has been made available at https://doi.org/10.1594/PANGAEA.862525. In this study, the accuracy of the ERA-Interim-derived surface pressure (with two computation methods) and GPT2w-derived pressure for 108 GNSS stations during the period 2000–2013 has been investigated by comparing them against the observed

pressure. The results indicate that ERA-Interim has the potential to be used as a meteorological data source for obtaining high accuracy PWV results on a global scale for climate study and the GPT2w can be potentially used for a similar application but may be only suitable to be used in the tropical regions.

The biases of both ERA-Interim-derived and GPT2w-derived pressures are between –1 and 1 mb at most stations, whilst GPT2w has a far better performance in the tropical regions than both mid and high latitude regions. For those stations

located in low latitude regions, the RMS errors of GPT2w-derived pressure are in the range of 1.0~4.3 mb with the mean of 2.4 mb. However, for the stations located in the mid to high latitude belts, the mean value of the RMS errors is 7.4 mb, and for the stations located in the high-latitude belts, the mean of the RMS errors is 9.8 mb. The RMS errors of the two sets of ERA-Interim-derived pressure (with different computation methods) at all the 108 stations are both in the range of 0.2~4.0 mb and the mean value is 1.1 mb.

The biases and RMS errors of the ZHDs based on ERA-Interim-derived and GPT2w-derived pressure are calculated by using the ZHD values based on the observed pressure as a reference. The biases of the three sets of ZHD are all in the range



of –3~3 mm at most stations. The RMS errors of the GPT2w-derived ZHD are in the range of 2.3~28.7 mm at all the 108 stations. For the stations located in the low-latitude belts, the RMS errors of the ZHDs are within 2.3~9.9 mm and the mean of the RMS errors is 5.6 mm, which is significantly smaller than that at the stations located in the other latitude belts. For the

two sets of ERA-Interim-derived ZHDs, their RMS errors are similar and all in the range of 0.5~9.1 mm with the mean of 2.5 mm. In addition, the RMS errors of these three sets of pressure and ZHD are obviously larger in the cold season than that in the warm season, especially the GPT2w-derived values.

To investigate the impact of the above error in the pressure results on the resultant PWV, the PWVs determined from the aforementioned three sets of pressure are compared against the PWV derived from the observed pressure. The ZTD provided

by CODE and the conversion factor computed using temperature and humidity profiles from ERA-Interim are adopted in the conversion from ZTD to PWV. The results show that the RMS errors of the GPT2w-derived PWVs are in the range of 0.5~4.2 mm. The mean value of the RMS errors of these PWVs is 1.0 mm for all the stations located within -30°~30°. However, for the stations located in the high-latitude belts, the mean value of the RMS errors is as large as 3.4 mm. The RMS errors of the ERA-Interim-derived PWV are all in the range of 0.1~1.5 mm with the mean of about 0.4 mm. The

investigation results also indicate that ERA-Interim-derived pressure can be used as a potential global useful meteorological data source for computing the PWV to achieve an acceptable accuracy for climate studies. However, the GPT2w-derived pressure may only be used in the tropical regions for the same purpose and not be suitable for other regions.

## Acknowledgement

This project is funded by The Natural Disaster Resilience Grants Scheme (NDRG) of Victoria, the Australian Research

Council (ARC) Linkage (LP0883288) projects funded by the Australian Federal Government, the China Scholarship Council (CSC), the National Natural Science Foundation of China (No. 41374014), and Service platform construction and demonstration of BDS geodetic datum (B1503). We thank the Center for Orbit Determination in Europe for providing the zenith total delay results, the European Centre for Medium-Range Weather Forecasts for providing the ERA-Interim reanalysis, and the Scripps Orbit and Permanent Array Center for providing the surface pressure measurements. We also

thank Böhm et al for providing the GPT2w model. We appreciate the PANGAEA to help us publish our new PWV dataset, especially the help from Dr. Stefanie Schumacher and Rainer Sieger.

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



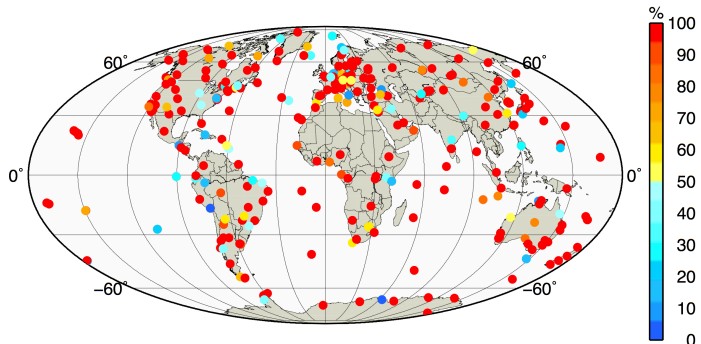

**Figure 1. Missing rate of the surface pressure observations compared to the GNSS-derived ZTD at 371 IGS stations for the period of 2000–2013**


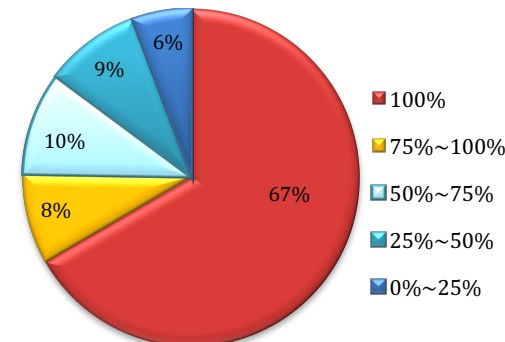

**Figure 2. Statistic results of the missing rates of the surface pressure observations across 371 stations for the period of 2000–2013**





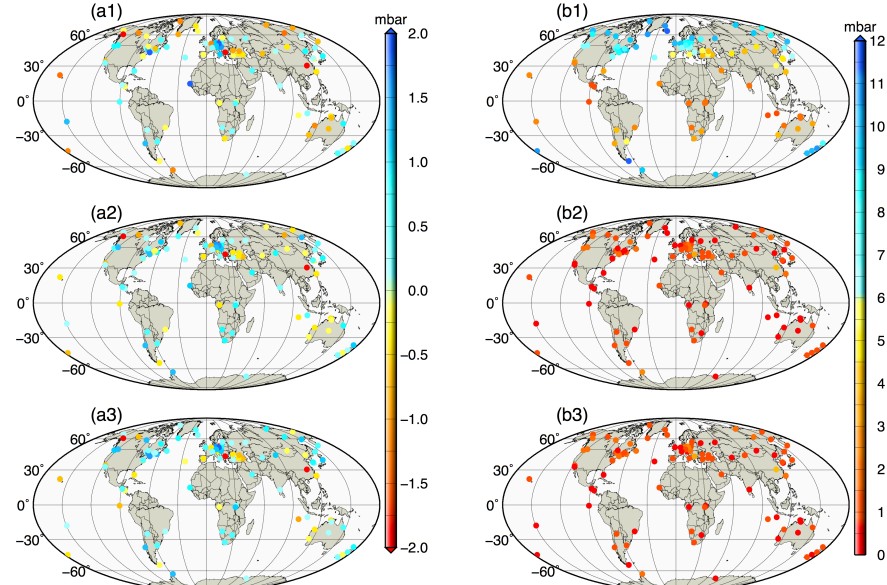

**Figure 3. Biases of *P_GPT2w* (a1), *P_ERA*1 (a2), *P_ERA*4 (a3) and the RMS errors of *P_GPT2w* (b1), *P_ERA*1 (b2) and *P_ERA*4 (b3) across 108 GNSS stations for the period of 2000–2013**

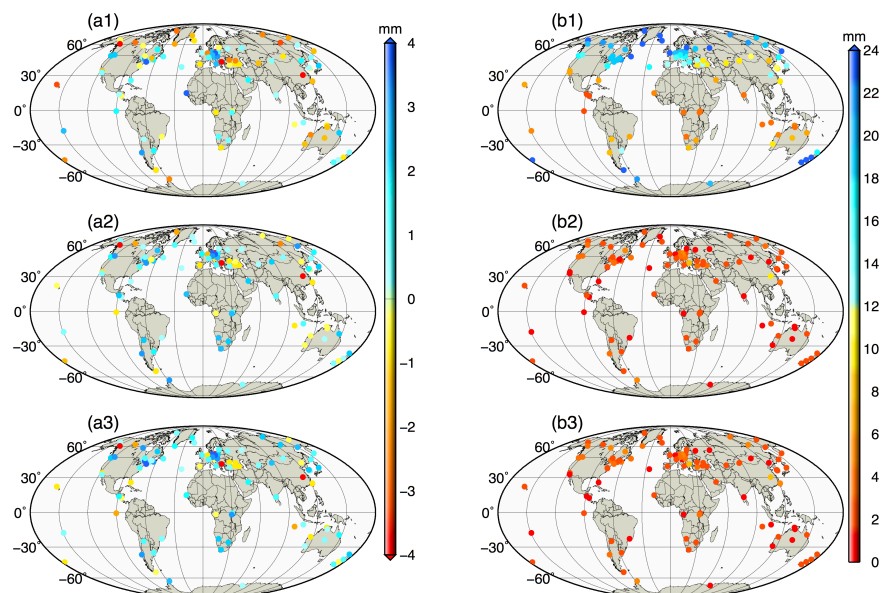

**Figure 4. Biases of the *ZHD_GPT2w* (a1), *ZHD_ERA*1 (a2), *ZHD_ERA*4 (a3) and RMS errors of *ZHD_GPT2w* (b1), *ZHD_ERA*1 (b2), *ZHD_ERA*4 (b3) across 108 GNSS stations for the period of 2000–2013**





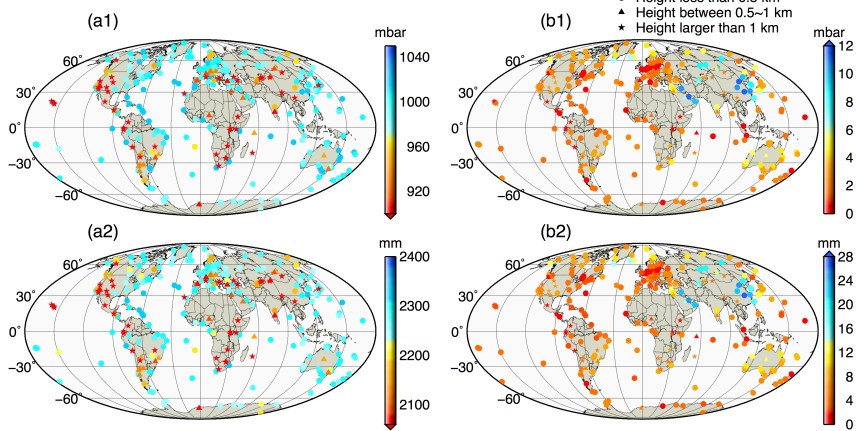

**Figure 5.** Mean values of the pressure (a1) and ZHD (a2) across 371 stations for the period 2000–2012; Annual amplitudes of the pressure (b1) and ZHD (b2) across 371 stations for the period 2000–2012


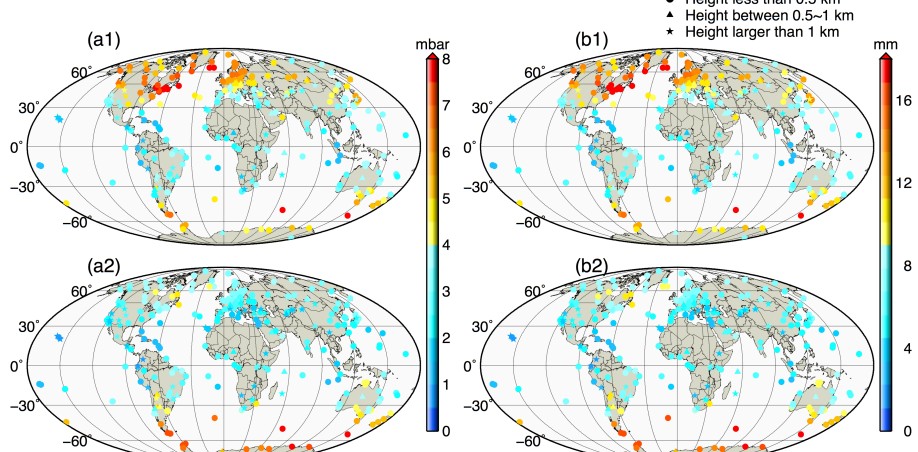

**Figure 6.** Diurnal range of the pressure across 371 stations for December–February (a1) and June–August (a2); Diurnal range of the ZHD across 371 stations for December–February (b1) and June–August (b2)





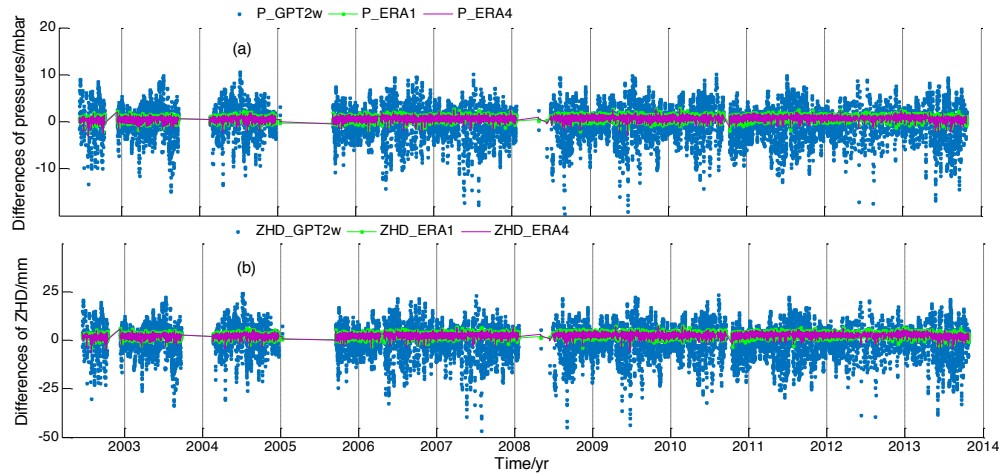

**Figure. 7 Time series of errors in pressure (a) and ZHD (b) for station SUTM located in the southern hemisphere**

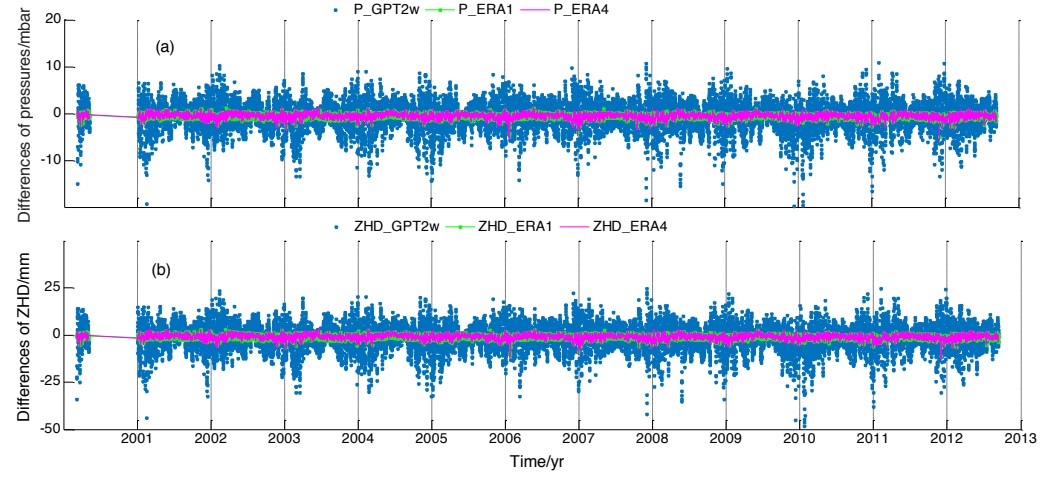

**Figure. 8 Time series of errors in pressure (a) and ZHD (b) for station JPLM located in the northern hemisphere**

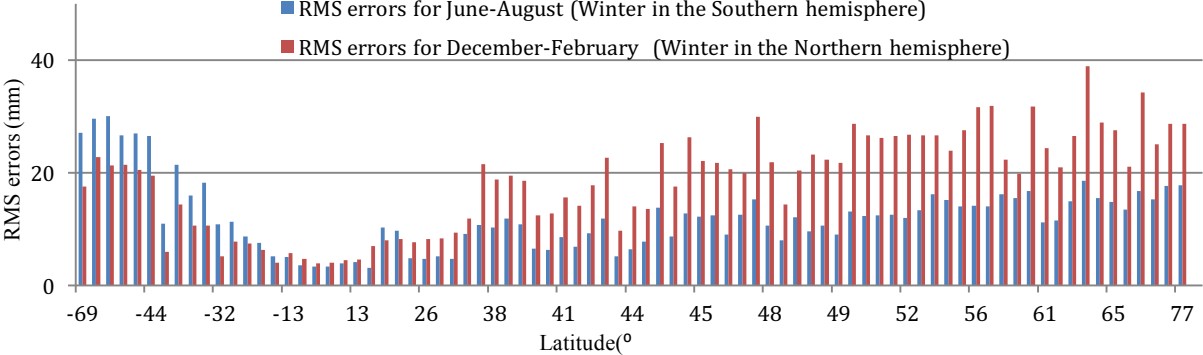

**Figure 9. RMS errors of GPT2w-derived ZHD for the months June–August and December–February**





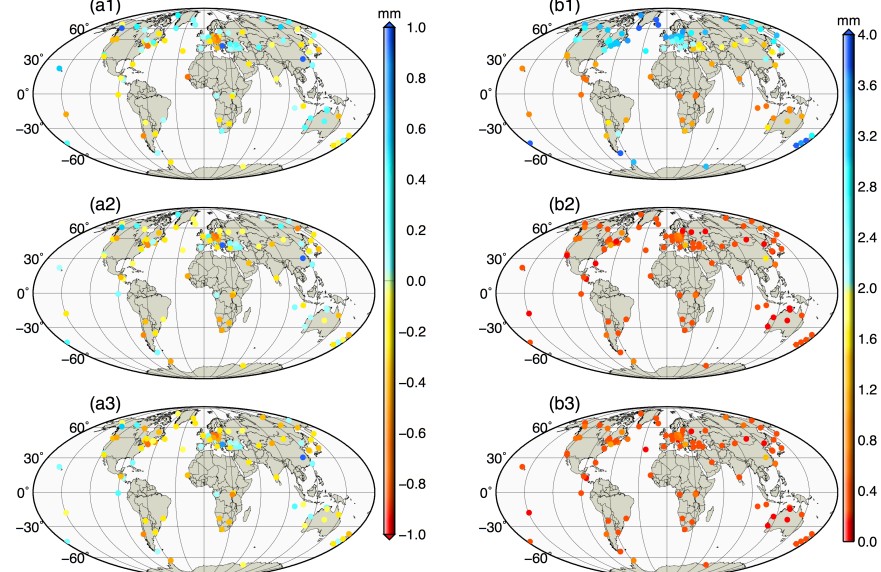


**Figure. 10 Biases of the *PWV_GPT*2*w* (a1), *PWV_ERA*1 (a2), *PWV_ERA*4 (a3) and RMS errors of *PWV_GPT*2*w* (b1), *PWV_ERA*1 (b2), *PWV_ERA*4 (b3) of 108 GNSS stations for the period of 2000–2013**