# Peer review of "Determination of zenith hydrostatic delays and the development of new global long-term GNSS-derived precipitable water vapor"

_Atmospheric Measurement Techniques, 2016_

## Referee Comment (RC1) · Anonymous Referee #1 · 13 Sep 2016

**Review of *Determination of zenith hydrostatic delays and the development of new global long-term GNSS-derived precipitable water vapor* by Wand, Zhang, Wu, He, Cheng and Li, AMT-2016-264**

**Overview**

To derive precipitable water (PW) from GNSS zenith total delay (ZTD) data, an estimate of the zenith hydrostatic delay (ZHD) is needed at the location of the GNSS reciever antenna in order to obtain the zenith wet delay (ZWD) from which PW can be deduced. ZHD is easily deduced given the location of the GNSS receiver antenna and the pressure at the same location.

This short, but interesting paper address the problem that often no pressure sensor data are available at GNSS sites and consider two alternative sources of pressure estimates. One is the empirical Global Pressure and Temperature 2 wet model (GPT2w), the other numerical weather prediction re-analysis data in the form of ERA-Interim data at 6 hourly time resolution, 80 km horisontal resolution and 60 vertical levels. Based on offset statistics for 108 GNSS sites at which pressure sensor data are available, it is concluded ERA data are best. This is not surprising, given GPT2w is derived from ERA data and neglects the daily variations, but here are numbers to detail it in various ways. Further it is concluded that of the two, only ERA data are of sufficient quality to provide PW estimates of the quality necessary for climate monitoring on global scale, whereas GPT2w might be used in the tropics.

Obviously a more throughrough analysis could have been made doing this type of analysis for all sites with available pressure sensor data of decent quality, without restriction to GNSS sites. However, the number and distribution of sites included in this analysis is high enough to merit the conclusions.

Overall the paper is worthy of publication. A number of small issues which ought to be improved are listed in the detailed comments below.

**Detailed comments**

There is a mismatch between use of numbers and symbols in equations, like 4 and 5 versus 14 and 15. I recommend use of symbols throughout. List in the text which values you used for the symbols when you cranked out numbers.

Regarding equation 16 $T_m$ is the inverse of the water vapour weighted inverse temperature. Is does not read well, but it is not the same as the water vapour weigthed mean of temperature. You also need to specify the coordinate system (pressure levels or height levels; is $Pv_i$ halfway between top and bottum of gridbox in pressure space or height space?).

An analysis is provided of the pressure offset statistics as function of the altitude of the sites. I recommend to show instead (or also) the offset statistics as function of the absolute offset between the model altitude (GPT2w or ERA) versus the GNSS antenna altitude. The higher such offsets, the more the short commings due to model resolution, and of the interpolation/extrapolation methods on page 3 and 4, can be expected to show up.

It is difficult to read the global maps properly. When zooming in on a computer screen, one realizes that a significant part of the sites appear in a few site dense regions, the overall color of which are dominated by the symbols plotted last, on top of the other ones. Some sites may even not be visible. When different symbols are used, fig 5 and 6, it becomes even more difficult.

Consider turning most of the global maps into figures of the same type as figure 16, which is easy to read on an A4 printout. As an alternative consider to plot on the global maps the average for the sites for the regions in which the sites are not visible individually on the global maps at present.

---

## Referee Comment (RC2) · Anonymous Referee #2 · 19 Sep 2016

Review of manuscript amt-2016-264 "Determination of zenith hydrostatic delays and the development of new global long-term GNSS-derived precipitable water vapor" by Xiaoming Wang and co-authors.

General comments

The conversion of GNSS ZTD to PWV requires the use of surface pressure data to estimate the hydrostatic delay component. Errors in the surface pressure add uncertainty in the PWV results and may lead to erroneous conclusions on climate variations. This manuscript investigates the accuracy of surface pressure data from two global datasets based on the ERA-Interim reanalysis: the GPT2w, a coarse spatial/temporal resolution (5° mean horizontal, with annual and semi-annual cycles) version of the reanalysis commonly used by geodesists for GNSS data processing, and the legacy reanalysis at high spatial/temporal resolution (80-km horizontal, 60 levels, 6-hourly). This topic is very important to the GNSS/climate community and the proposed study is pertinent to the AMT journal aims and scope. However, the approach followed by the authors needs substantial improvement and the results need be analysed more thoroughly to be really useful to the scientific community. Given the importance of the topic and work already done by the authors, my suggestion is a major revision. I give below the main issues which should be solved and improvements that should be brought to the organisation of results before publication.

Major comments

1. On the quality of the reference pressure observations

The accuracy of the reanalysis data is evaluated with respect to surface pressure observations available from the IGS and distributed by SOPAC. Nothing is said about the accuracy of the IGS data in the manuscript. Have these data been quality controlled? How can their accuracy be established to be suitable for serving as a reference at the level of climate requirements? It has been shown in past studies (Wang et al., 2007; Heise et al., 2009) that the IGS meteorological data are generally of poor quality. I urge the authors either to use another, validated, surface pressure dataset (e.g. ISPD, see the work of Lagler et al., 2013), or to thoroughly screen the IGS data and select a subset of high quality stations. The fact that many biases detected in the GPT2w data appear also in the ERA-Interim data at much higher spatial resolution (Fig. 3a-c) suggest that these biases might in fact be in the IGS pressure data.

2. On the interpolation methods for ERA-Interim data

Two methods are introduced in section 3 for interpolating the ERA-Interim data from the model grid to the GNSS site. The first one is based on the nearest grid point and the second one is using the 4 surrounding grid points. The motivation for comparing these two methods should be better explained and their results should be discussed and interpreted in a more comprehensive way. The impact of representativeness errors should also be discussed when comparing model data and observations. However, in its present form, I suspect a major issue in the results due an inconsistency between the vertical interpolations used in both methods. Whereas the first method is based on the standard formula – eq (1) - assuming and constant lapse rate (linear temperature variation with height in the troposphere), the second one follows Schüler, 2001, and uses an empirical formula – eq (4) or (5) – which is inconsistent with eq (1) and poorly validated for usage at global scale. Moreover,

the weighted interpolation from 2 model levels – eq (3), (6) and (7) is a commonly used approach for horizontal interpolation but is not a priori valid for vertical interpolation because it would not conserve mass (vertical pressure variations should satisfy hydrostatic equilibrium). Tracking back the origin and validity of these equations in Schüler, 2001, their usage for climate purposes appears highly questionable. I urge the author either to demonstrate in an appendix the validity of these equations at global scale or bring the vertical interpolation in line with the first method.

3. Objectives of the work and interpretation of the results

Though it is a priori obvious that GPT2w will give worse results than Era-Interim due to the difference in spatial and temporal resolutions, quantifying the spatial distribution of errors and decomposing them into different time scales (mean, seasonal, diurnal) is useful in an assessment study. In this respect, the Introduction should better state the overall aim of this study and introduce the requirements in terms of accuracy on the studied data for climate applications. Once the target accuracy is specified it is easier to conclude on the observed results. The reference to the E-GVAP Product Reference Document given P8 should thus be provided in the Introduction. Note however, that the E-GVAP requirements may not be adequate for global climate as they are only expressed in a single value in kg m-2 unit. Therefore the requirements should be complemented with GCOS recommendations and expressed either in % or consider different values in different climate zones.

Tables presenting results in latitude bands and plots of results as a function of latitude might be useful to give a synthetic and more legible view than the hard to read plots (Fig. 3 and similar) and lengthy and repetitive descriptions in the text (similar results for pressure, ZHD, and PWV).

The spatial distribution and temporal variations of pressure/ZHD (Fig. 5-8) are well known climatic features (e.g. Trenberth, 1981; Dai and Wang, 1999). The text and comments should be revised accordingly.

Trenberth, K. E. (1981), Seasonal variations in global sea level pressure and the total mass of the atmosphere, J. Geophys. Res., 86(C6), 5238–5246, doi:10.1029/JC086iC06p05238.

Dai, A., & Wang, J. (1999). Diurnal and semidiurnal tides in global surface pressure fields. Journal of the atmospheric sciences, 56(22), 3874-3891.

The ZTD data introduced in section 4.4 are not used in fact because the error in PWV due to surface pressure does not depend on ZTD but only on ZHD and the conversion factor PI. So the ZTD could be completely avoided in this study unless the relative PWV errors are computed, in which case the results would depend on ZHD and ZTD (and no longer on PI). I suggest that the authors present also the relative PWV errors which might also highlight shortcomings in the Polar Regions.

The authors conclude that ERA-Interim pressure data can be used globally for climate studies while GPT2w may be suitable only in the tropics. These conclusions are simply based on the E-GVAP thresholds and the results obtained from the comparison of 6-hourly data. However, it is obvious that for climate applications, it might often be sufficient to consider monthly means. Hence the random errors would be reduced accordingly and a larger number of sites might be considered. This study should thus provide also results for monthly mean data.

At the end of section 4.4.1, it is written that ERA-Interim data yield RMS errors < 0.5mm at 75 or 78% of the sites. What happens at the remaining 22 or 25%? Should these stations be blacklisted?

The discussion and conclusion must also take into account the presence of systematic errors.

4. On the presentation of results

In section 4 of the manuscript, the results for surface pressure, ZHD, and PWV, are presented successively. In each case, the biases and RMS errors characterizing the surface pressure difference between the tested model and the reference observations are presented. As attested by eq (11) and (13), an error in surface pressure produces a proportional error in ZHD and PWV which can be quantified almost exactly by the rule of thumb: 2.3 mm/hPa and 1 kg m-2 / 6.5 mm, respectively. As a consequence, the spatial distributions of biases and RMS errors presented in Fig. 4 and 10 are quasi similar to those shown in Fig. 3 and don't add information. This is also the case for Fig. 5 -8 (pressure and ZHD). I suggest that the authors combine the results in one figure when possible and add data axis (or colorbars) with multiple scales for pressure, ZHD, and PWV. This would avoid unnecessary duplication of figures and leave room for additional information.

Minor comments

- The IS unit for pressure is hPa (not mbar)
- The preferred unit for PWV is kg m-2 as mm may be mixed up with the ZHD unit.
- It is written P3L87 that the ERA-Interim data are available on 60 model levels, but later the equations referring to computed quantities refer to pressure levels (section 3.2 and 4). Please clarify.
- Section 4.3: it is not said which of the two ERA-Interim datasets is used.
- Section 4.3: it is not said how the annual and semi-annual amplitudes are computed.
- The information provided in Fig. 2 might be simply added in the captions of Fig. 1.
- Fig. 3 and alike are too small to be useful. Consider using full page width.
- P9L245: add a reference to the E-GVAP Product Reference Document rather than citing the website (http://egvap.dmi.dk/)
- Reference to Yao et al., GJI, 2014, is not complete and might be replaced with a more recent paper by Yao et al., Science China, 2015.

---

## Author Comment (AC1) · 4 Nov 2016

The comment was uploaded in the form of a supplement:
http://www.atmos-meas-tech-discuss.net/amt-2016-264/amt-2016-264-AC1-supplement.pdf

---

## Author Comment (AC2) · 4 Nov 2016

**Review of manuscript amt-2016- development of new global long-term GNSS-derived precipitable water vapor" by Xiaoming Wang and co-authors.**

**General comments**

*The conversion of GNSS ZTD to IWV requires the use of surface pressure data to estimate the hydrostatic delay component. Errors in the surface pressure add uncertainty in the IWV results and may lead to erroneous conclusions on climate variations. This manuscript investigates the accuracy of surface pressure data from two global datasets based on the ERA-Interim reanalysis: the GPT2w, a coarse spatial/temporal resolution (5° mean horizontal, with annual and semi-annual cycles) version of the reanalysis commonly used by geodesists for GNSS data processing, and the legacy reanalysis at high spatial/temporal resolution (80-km horizontal, 60 levels, 6-hourly). This topic is very important to the GNSS/climate community and the proposed study is pertinent to the AMT journal aims and scope. However, the approach followed by the authors needs substantial improvement and the results need be analyzed more thoroughly to be really useful to the scientific community. Given the importance of the topic and work already done by the authors, my suggestion is a major revision. I give below the main issues which should be solved and improvements that should be brought to the organization of results before publication.*

**Response:** We appreciate the comments from the reviewer, which are very constructive and helpful. All these constructive comments have been taken into consideration to further improve our manuscript. New results and discussions regarding the application of GNSS-derived monthly IWV for climate studies have been added in the revised manuscript. The structure of the manuscript has been changed based on the reviewer's suggestion to make it more logical. All issues raised by the reviewer have been addressed thoroughly and the following is our response to each of the questions from reviewer #2.

**Major comments**

1. *On the quality of the reference pressure observations: The accuracy of the reanalysis data is evaluated with respect to surface pressure observations available from the IGS and distributed by SOPAC. Nothing is said about the accuracy of the IGS data in the manuscript. Have these data been quality controlled? How can their accuracy be established to be suitable for serving as a reference at the level of climate requirements? It has been shown in past studies (Wang et al., 2007; Heise et al., 2009) that the IGS meteorological data are generally of poor quality. I urge the authors either to use another,*

*validated, surface pressure dataset (e.g. ISPD, see the work of Lagler et al., 2013), or to thoroughly screen the IGS data and select a subset of high quality stations. The fact that many biases detected in the GPT2w data appear also in the ERA- Interim data at much higher spatial resolution (Fig. 3a-c) suggest that these biases might in fact be in the IGS pressure data.*

**Response:** As suggested by previous studies (Bianchi et al., 2016; Heise et al., 2009; Wang et al., 2007), meteorological data provided by IGS need to be rigorously screened before they are used to calculate ZHD. In the revised version of manuscript, a quality control process has been performed on surface pressure observations used in this study. The procedure of the quality control process has been also added to L91−L100 as: "*Surface pressure provided by SOPAC is used to validate the performance of ERA-Interim and GPT2w derived surface pressure. However, as suggested by previous studies (Bianchi et al., 2016; Heise et al., 2009; Wang et al., 2007), meteorological data provided by IGS need to be rigorously screened before they are used to calculate IWV. In this study, the pressure values from 131 stations are screened carefully to prevent those surface pressure observations with a poor quality being used. Firstly, the time series of pressure at all 131 stations are checked carefully to delete stations at which the pressure values have obvious large noises or offsets (might be caused by the change of pressure sensors). This leads to 23 stations are deleted and the remaining 108 stations are used in this study. Then the pressure values at these 108 stations are further checked for detecting and excluding unrealistic values out of the range between 550 and 1100 hPa. The third and also the last step is identifying those pressure values that depart from the mean value with more than three standard deviations at each station, which leads to about 0.5 % of the data are detected and excluded from use.*"

However, after the above quality control mechanisms are in place (L189−191), "*there are still 8 stations (JOZ2, WHIT, WROC, OHIG, GOPE, BOR1, SOFI and WUHN) that have a RMS error larger than 2 hPa. The large difference between the surface pressure observations and the pressure derived from ERA-Interim is probably caused by poor quality observations at these stations.*" More details about the RMS errors and biases of pressures, ZHD and IWV have been provided in our supplementary documents including these 8 stations, which have been labelled with the red colour.

2. *On the interpolation methods for ERA-Interim data: Two methods are introduced in section 3 for interpolating the ERA-Interim data from the model grid to the GNSS site. The first one is based on the nearest grid point and the second one is using the 4 surrounding grid points. The motivation for comparing these two methods should be better explained and their results should be discussed and interpreted in a more comprehensive way. The impact of representativeness errors should also be discussed when comparing model data and observations. However, in its present form, I suspect a major issue in the results due an inconsistency between the vertical interpolations used in both methods. Whereas the first method is based on the standard formula – eq (1) - assuming and constant lapse rate*

*(linear temperature variation with height in the troposphere), the second one follows Schüler, 2001, and uses an empirical formula – eq (4) or (5) – which is inconsistent with eq (1) and poorly validated for usage at global scale. Moreover, the weighted interpolation from 2 model levels – eq (3), (6) and (7) is a commonly used approach for horizontal interpolation but is not a priori valid for vertical interpolation because it would not conserve mass (vertical pressure variations should satisfy hydrostatic equilibrium). Tracking back the origin and validity of these equations in Schüler, 2001, their usage for climate purposes appears highly questionable. I urge the author either to demonstrate in an appendix the validity of these equations at global scale or bring the vertical interpolation in line with the first method.*

**Response:** Results from these two methods are compared on a global scale to investigate whether there is any difference between them. As suggested by the reviewer, a discussion on the representativeness error when comparing model data and point observations has been added (L189–201): "*However, there are still 8 stations (JOZ2, WHIT, WROC, OHIG, GOPE, BOR1, SOFI and WUHN) that have a RMS error larger than 2 hPa. The large difference between the surface pressure observations and the pressure derived from ERA-Interim is probably caused by poor quality observations at these stations. Another possible reason for this large difference is the representativeness error in the ERA-Interim model due to the limited model resolution (Janjic and Cohn, 2006;Buehler et al., 2012;Waller et al., 2014). The representativeness error arises when the point observations can represent small spatial scales well but not the model and this error can be expected extreme in complex mountainous terrain, where there is a mismatch between the model and actual terrain (Jiménez and Dudhia, 2012;Ancell et al., 2011;Zhang et al., 2013;Duan et al., 2015). For example, stations WROC and GOPE that have large RMS errors are both located in the Sudety Mountains, Poland, where the atmospheric variables have rapid and frequent changes. The pressure determined using either one-point method or four-point method did not have a good accuracy. In this study, we cannot conclude whether this large difference is caused by either the poor quality of the surface pressure or the representativeness error in ERA-Interim. More studies are needed for those regions with a complex terrain using accurate and reliable local meteorological observations.*"

The vertical interpolation procedure for the four-point method has been changed to the same with the first method and all the results and comparison in the revised version of manuscript have been presented based on the new results.

3. *Objectives of the work and interpretation of the results: Though it is a priori obvious that GPT2w will give worse results than Era-Interim due to the difference in spatial and temporal resolutions, quantifying the spatial distribution of errors and decomposing them into different time scales (mean, seasonal, diurnal) is useful in an assessment study. In this respect, the Introduction should better state the overall aim of this study and introduce the requirements in terms of accuracy on the studied data for climate applications. Once the*

*target accuracy is specified it is easier to conclude on the observed results. The reference to the E-GVAP Product Reference Document given P8 should thus be provided in the Introduction. Note however, that the E-GVAP requirements may not be adequate for global climate as they are only expressed in a single value in kg m-2 unit. Therefore, the requirements should be complemented with GCOS recommendations and expressed either in % or consider different values in different climate zones.*

**Response:** The requirement of the GNSS-derived IWV for climate studies has been added in the introduction section in terms of both absolute accuracy (kg m$^{-2}$) and relative accuracy (%) (L62−67): "*As stated in the product requirement document from the EIG EUMETNET GNSS water vapor programme (Offiler, 2010), the accuracy of IWV for Global Climate Observing System (GCOS) required is better than 3 kg m$^{-2}$. The "breakthrough" and "goal'' accuracy is 1.5 kg m$^{-2}$ and 1 kg m$^{-2}$, respectively. Bock et al (2013) also pointed out that for climate monitoring it needs to achieve a 3 % or better accuracy from GNSS-derived IWV. Since for climate studies, the accuracy of IWV (in terms of RMS) can be obtained by averaging observations of a site over a long period (e.g. a month), the error in IWV is studied not only for the four epochs of each day but also for the monthly mean.*"

4. *Tables presenting results in latitude bands and plots of results as a function of latitude might be useful to give a synthetic and more legible view than the hard to read plots (Fig. 3 and similar) and lengthy and repetitive descriptions in the text (similar results for pressure, ZHD, and IWV). The spatial distribution and temporal variations of pressure/ZHD (Fig. 5-8) are well known climatic features (e.g. Trenberth, 1981; Dai and Wang, 1999). The text and comments should be revised accordingly.*

   - *Trenberth, K. E. (1981), Seasonal variations in global sea level pressure and the total mass of the atmosphere, J. Geophys. Res., 86(C6), 5238–5246, doi:10.1029/JC086iC06p05238.*
   - *Dai, A., & Wang, J. (1999). Diurnal and semidiurnal tides in global surface pressure fields. Journal of the atmospheric sciences, 56(22), 3874-3891.*

**Response:** A new table about the comparison results of pressure, ZHD and IWV has been added to the new manuscript as suggested by the reviewer. New references have been added in the manuscript about the spatial and temporal variations of pressure: "*Previous studies have indicated that global surface pressure undergoes annual, semi-annual and diurnal variations (Dai and Wang, 1999; Trenberth, 1981).*"

Tab. 1 The mean values of the biases and RMSs of pressure, ZHD and IWV derived from three methods for the stations in low-, mid- and high-latitude regions

| Method | Region | Pressure (hPa) | | ZHD (mm) | | IWV (kg m$^{-2}$) | |
|---|---|---|---|---|---|---|---|
| | | Bias | RMS | Bias | RMS | Bias | RMS |
| GPT2w | Low-latitude | 0.06 | 2.35 | 0.13 | 5.36 | −0.03 | 0.94 |
| | Mid-latitude | 0.21 | 7.16 | 0.47 | 16.33 | −0.07 | 2.57 |
| | High-latitude | −0.74 | 9.60 | −1.69 | 21.80 | 0.23 | 3.29 |
| ERA-Interim (One-point) | Low-latitude | 0.17 | 0.69 | 0.69 | 1.72 | −0.11 | 0.30 |
| | Mid-latitude | 0.23 | 1.16 | 0.73 | 2.69 | −0.11 | 0.42 |
| | High-latitude | −0.19 | 1.10 | −0.31 | 2.45 | 0.04 | 0.37 |
| ERA-Interim (Four-pint) | Low-latitude | 0.13 | 0.71 | 0.30 | 1.62 | −0.05 | 0.28 |
| | Mid-latitude | 0.23 | 1.13 | 0.54 | 2.58 | −0.08 | 0.41 |
| | High-latitude | −0.19 | 1.04 | −0.44 | 2.37 | 0.06 | 0.36 |

5. *The ZTD data introduced in section 4.4 are not used in fact because the error in IWV due to surface pressure does not depend on ZTD but only on ZHD and the conversion factor PI. So the ZTD could be completely avoided in this study unless the relative IWV errors are computed, in which case the results would depend on ZHD and ZTD (and no longer on PI). I suggest that the authors present also the relative IWV errors which might also highlight shortcomings in the Polar Regions. The authors conclude that ERA-Interim pressure data can be used globally for climate studies while GPT2w may be suitable only in the tropics. These conclusions are simply based on the E-GVAP thresholds and the results obtained from the comparison of 6-hourly data. However, it is obvious that for climate applications, it might often be sufficient to consider monthly means. Hence the random errors would be reduced accordingly and a larger number of sites might be considered. This study should thus provide also results for monthly mean data. At the end of section 4.4.1, it is written that ERA-Interim data yield RMS errors < 0.5mm at 75 or 78% of the sites. What happens at the remaining 22 or 25%? Should these stations be blacklisted?. The discussion and conclusion must also take into account the presence of systematic errors.*

**Response:** In the revised manuscript, a new section titled "**GNSS-derived IWV for climate studies**" has been added to study the possibility of using GNSS-derived monthly IWV for climate studies. In this section, the difference between the monthly IWV obtained using pressure from ERA-Interim and GPT2w are compared with the monthly IWV obtained using observed pressure in terms of both absolute error and relative error, see (L255–L276): "*As suggested by the E-GVAP, the "breakthrough" accuracy of IWV for climate study is 1.5 kg m$^{-2}$ and the "goal" accuracy of IWV for climate study is around 1 kg m$^{-2}$. Bock et al (2013) also stated that GNSS-derived IWV with an accuracy of 3 % or better is sufficient for climate*

*monitoring. Since for climate studies, the accuracy of IWV (in terms of RMS) can be obtained by averaging all observations of a site over a long period (e.g. a month). The accuraces of the monthly IWV resulted from GPT2w and ERA-Interim derived pressure are measured by the difference from the monthly IWV resulted from surface pressure observations (the latter is the reference). IWVs obtained from surface pressure P_GPT2w, P_ERA1, and P_ERA4 are named as IWV_GPT2w, IWV_ERA1, and IWV_ERA4, respectively. Tab. 2 shows the statistic result of the bias, RMS and relative error of the monthly IWV derived from the aforementioned three methods at 98 stations. It should be noted that the aforementioned eight stations with possible poor data quality and two stations with large data missing rate and thus cannot get a mean result are not used in Tab. 2. It can be found that the error in both IWV_ERA1 and IWV_ERA4 is quite small, with a RMS error of about 0.2 kg m$^{-2}$ on a global scale. The relative errors in both IWV_ERA1 and IWV_ERA4 in the low-latitude, mid-latitude and high-latitude regions are about 0.8 %, 1.8 % and 2.5 %, respectively. The mean relative error of IWV_ERA1 and IWV_ERA4 across all 98 stations is about 1.4 %. However, for IWV_GPT2w, the relative error is as large as 6.7 % in the mid-latitude regions and can be up to 21.5 % in the polar regions. Therefore, the monthly IWV resulted from ERA_Interim-derived pressure has a good accuracy, especially in the low- and mid-latitude regions, thus it has the potential to be used for climate studies. However, the accuracy of the monthly IWV resulting from GPT2w-derived pressure is not good enough for climate studies in the mid- and high-latitude regions*".

Tab. 2 Bias, RMS and relative error of monthly IWVs derived from three methods

| Method | Region | Bias (kg m$^{-2}$) | RMS (kg m$^{-2}$) | Relative error (%) |
|---|---|---|---|---|
| GPT2w | Low-latitude | 0.09 | 0.38 | 1.52 |
| | Mid-latitude | 0.10 | 0.92 | 6.70 |
| | High-latitude | −0.25 | 1.44 | 21.48 |
| ERA-Interim (One-point) | Low-latitude | 0.11 | 0.22 | 0.91 |
| | Mid-latitude | 0.10 | 0.25 | 1.80 |
| | High-latitude | −0.03 | 0.17 | 2.63 |
| ERA-Interim (Four-pint) | Low-latitude | 0.05 | 0.19 | 0.72 |
| | Mid-latitude | 0.07 | 0.24 | 1.70 |
| | High-latitude | −0.04 | 0.16 | 2.39 |

We have added two supplementary documentations containing the bias and RMS error of each station at four epochs of each day and also for the monthly mean of IWV. This can help readers to decide whether the pressure derived from ERA-Interim and GPT2w at a station is accurate enough for their studies or applications.

As suggested by the reviewer, a discussion regarding the systematic errors in the pressure derived from ERA-Interim and GPT2w: "*Fig. 8 shows the scatter plot between the IWV derived from surface pressure (X-axis), and IWV_GPT2w, IWV_ERA1, and IWV_ERA4 (Y-*

*axis) at 9 stations. As shown in this figure, both IWV_ERA1 and IWV_ERA4 do not contain obvious biases compared to the IWV derived from surface pressure observations. However, for the IWV_GPT2w, obvious biases can be found at several stations and the characteristics of biases are different at different stations."*

[Figure]

Figure 8 Scatter plot of the IWV determined using surface pressure observations (X-axis) and *IWV_GPT2w*, *IWV_ERA1*, and *IWV_ERA4 (*Y-axis*)*

6. *On the presentation of results: In section 4 of the manuscript, the results for surface pressure, ZHD, and IWV, are presented successively. In each case, the biases and RMS errors characterizing the surface pressure difference between the tested model and the reference observations are presented. As attested by eq (11) and (13), an error in surface pressure produces a proportional error in ZHD and IWV which can be quantified almost exactly by the rule of thumb: 2.3 mm/hPa and 1 kg m-2 / 6.5 mm, respectively. As a consequence, the spatial distributions of biases and RMS errors presented in Fig. 4 and 10 are quasi similar to those shown in Fig. 3 and don't add information. This is also the case for Fig. 5 -8 (pressure and ZHD). I suggest that the authors combine the results in one figure when possible and add data axis (or colorbars) with multiple scales for pressure, ZHD, and IWV. This would avoid unnecessary duplication of figures and leave room for additional information.*

**Response:** The biases and RMS errors of ZHD and IWV have been deleted and a statement on the relationship between the error in pressure and its resultant ZHD/IWV has been added: "*In terms of RMS, a 1 hPa error in the determined pressure will lead to a 2.3 mm error in its resultant ZHD and about a 0.38 kg m$^{-2}$ error in its resultant IWV. Therefore, the*

*characteristics of the spatial distribution of the errors in ZHD and IWV are quite similar to that in pressure*". Supplementary documentations for the biases and RMS errors at all stations have been also added.

**Minor comments**

*1. The IS unit for pressure is hPa (not mbar)*

**Response**: Amended

*2. The preferred unit for IWV is kg m-2 as mm may be mixed up with the ZHD unit.*

**Response**: Amended

*3. It is written P3L87 that the ERA-Interim data are available on 60 model levels, but later the equations referring to computed quantities refer to pressure levels (section 3.2 and 4). Please clarify.*

**Response**: it has been changed to "*Its spatial resolution is approximately 80 km in the horizontal direction and at 37 vertical pressure levels*"

*4. Section 4.3: it is not said which of the two ERA-Interim datasets is used and it is not said how the annual and semi-annual amplitudes are computed.*

**Response**: It has been clarified as "*In this section, annual amplitudes and diurnal ranges of pressure and ZHD over 371 stations for the period 2000–2013 are estimated with least squares under the assumption that there are only annual and semi-annual variations in the time series. The pressure and ZHD results derived from ERA-Interim with the four-point method are adopted for the estimation of amplitudes of the annual and semi-annual variations.*"

*5. The information provided in Fig. 2 might be simply added in the captions of Fig. 1.*

**Response:** Amended, Fig. 2 has been deleted and relevant information has been added in the caption of Fig. 1

*6. Fig. 3 and alike are too small to be useful. Consider using full page width.*

**Response**: All the figures have been enlarged.

*7. P9L245: add a reference to the E-GVAP Product Reference Document rather than citing the website (http://egvap.dmi.dk/)*

**Response:** A reference for the E-GVAP Product Reference Document has been added

8. *Reference to Yao et al., GJI, 2014, is not complete and might be replaced with a more recent paper by Yao et al., Science China, 2015.*

**Response:** The reference has been updated to date.